

**Homogenized daily sunshine duration over China from 1961 to 2022**
Yanyi He[1,3], Kaicun Wang[2*], Kun Yang[3], Chunlüe Zhou[1*], Changkun Shao[3], and Changjian Yin[2]
[1] School of Geography and Planning, Sun Yat-sen University, 510275 Guangzhou, China
[2] Institute of Carbon Neutrality, Sino-French Institute for Earth System Science, College of
Urban and Environmental Sciences, Peking University, 100081 Beijing, China
[3] Department of Earth System Science, Ministry of Education Key Laboratory for Earth System
Modeling, Institute for Global Change Studies, Tsinghua University, 100084 Beijing, China
[*]**Corresponding Author:** Kaicun Wang, kcwang@pku.edu.cn; Chunlüe Zhou,
zhouchunlue@mail.sysu.edu.cn

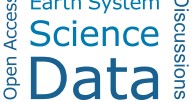

**Abstract**
Inhomogeneities in the sunshine duration (SSD) observational series, caused by non-climatic
factors like China's widespread transition from manual to automatic SSD recorders in 2019 or
station relocations, have hindered accurate estimate of near-surface solar radiation for the
analysis of global dimming and brightening as well as related applications, such as solar energy
planning and agriculture management. This study compiled raw SSD observational data from
1961 to 2022 at more than 2,200 stations in China and clearly found that the improved precision
from 0.1 hour to 1 minute following the instrument update in 2019 led to a sudden reduction
in the frequency of zero SSD from 2019 onwards, referred to as the day0-type discontinuity.
For the first time, we systematically corrected this known day0-type discontinuity at 378
stations (17%) in China, resulting in an SSD series with comparable frequencies of zero value
before and after 2019. On this base, we constructed a homogenization procedure to detect and
adjust discontinuities in both the variance and mean of daily SSD from 1961 to 2022. Results
show that a total of 1,363 (60%) stations experienced breakpoints in SSD, of which ~65% were
confirmed by station relocations and instrument replacements. Compared to the raw SSD, the
homogenized SSD is more continuous to the naked eye for various periods, and presents
weakened dimming across China from 1961 to 1990 but a non-significant positive trend by a
reduction of 60% in the Tibetan Plateau, suggesting that the homogenized SSD tends to better
capture the dimming phenomenon. The northern regions continue dimming from 1991 to 2022
but the southern regions of China brighten slightly. The implementation of the Action Plan for
Air Pollution Prevention and Control since 2013 has contributed to a reversal of SSD trend
thereafter, which is better reflected in the homogenized SSD with a trend shift from -0.02 to
0.07 hours·day$^{-1}$/decade from 2013 to 2022 in China, especially in heavily polluted regions.
Besides, the relationships of cloud cover fraction and aerosol optical depth with SSD are
intensified in the homogenized dataset. These results highlight the importance of the
homogenized SSD in accurately understanding the dimming and brightening phenomena. The
homogenized SSD dataset is publicly available for community use at
https://doi.org/10.11888/Atmos.tpdc.301478 (He et al., 2024).





## 1. Introduction


Sunshine duration (SSD) is one of the indispensable observation indicators in ground-
based meteorological measurements, capturing the duration of direct sunlight reaching the
Earth's surface (Wild et al., 2009; He et al., 2018). As an essential reference indicator to explore
variations in surface incident solar radiation ($R_s$), SSD has profound implications for
monitoring climate change, weather forecasting, ecosystem management, and solar energy
generation (Stanhill and Cohen, 2003; Baumgartner et al., 2018). Therefore, making high-
quality homogenized SSD data publicly accessible to diverse industries is crucial for research,
decision-making, and planning across various sectors.
SSD measurement dates back 170 years ago, when the sum of sub-periods for which direct
solar radiation exceeds 120 W/m$^2$ was defined as SSD (WMO, 2014). SSD measurements can
be broadly categorized into manual and automatic SSD recorders according to the need of
human supervision (Wang et al., 2021). The commonly used manual SSD recorders include the
Campbell-Stokes sunshine recorder and the Jordan sunshine recorder (Baumgartner et al.,
2018). These instruments operate by focusing direct solar radiation onto the photosensitized
paper, which burns and leaves one or more continuous traces that represents one or multiple
subperiods of sunshine duration (Che et al., 2005; Zhao et al., 2010). SSD is calculated as the
sum of the subperiods of the burn within a calendar day. Automatic sunshine recorders employ
sensors for observations and the types are diverse, including pyrheliometer, pyranometer,
photovoltaic sunshine recorders, and more (Lv et al., 2015).
Since the 1950s, the Jordan sunshine recorder has been the primary instrument for
measuring SSD in most meteorological stations in China. As reported, only 18 stations in the
Heilongjiang Province of Northeast China utilized the Campbell-Stokes sunshine recorder that
was subsequently replaced by the Jordan sunshine recorder in 2012 (Lu et al., 2012). In 2019,
China carried out a widespread replacement of the Jordon sunshine recorders transitioning to
the photoelectric digital SSD recorders at more than 2,400 stations to achieve the automation
of SSD measurement (Wang et al., 2020). In the first half of 2019, parallel observations were
conducted using both instruments, but starting from the second half of the year, only automatic
sunshine recorders were used to record SSD. Compared to traditional manual methods,





automatic sunshine recorders have higher precision and automation (Wang et al., 2020).
Recent studies have compared parallel observations for the two measurements at some
stations for certain regions of China (Lv et al., 2015; Hu et al., 2019; Lu et al., 2019; Lang et
al., 2021; Zhou et al., 2021b; Dai et al., 2022). They reported a relatively strong consistency
between both observations, but a certain degree of discrepancies still remain, which is closely
tied to the position of the sun and varying weather conditions: 1) the photoelectric digital
recorders tend to record higher values during weak direct radiation at sunrise and sunset and
lower values during strong noon radiation, compared to manual observations; and 2) under
persistently sunny weather, the more sensitive photoelectric digital recorders have slightly
longer SSD than the manual recorders, but the manual recorders under cloudy conditions with
intermittent sunshine tend to register artificially higher values due to lower instrument accuracy
and the spot effect.
Despite the absence of a sensitivity drift issue in SSD observations by manual sunshine
recorders, attributed to the daily replacement of photosensitized paper (Sanchez-Lorenzo and
Wild, 2012), the observational data still face challenges in ensuring consistency due to the
subjectivity introduced by different observers in practice. On the other hand, due to the
limitations of current observation technology, the photoelectric digital SSD recorders also have
shortcomings, such as narrow spectral response range, high sensitivity to nearby environment,
complex instrument maintenance, and instrument sensitivity drift and difficulty in calibration
(Wang et al., 2015; Wang et al., 2021). Several studies have confirmed that the replacement of
instruments can lead to non-climatic shifts in SSD and also applied a homogenization to SSD
in Iberian Peninsula (Sanchez-Lorenzo et al., 2007), Switzerland (Sanchez-Lorenzo and Wild,
2012), Italy (Manara et al., 2015), and Japan (Ma et al., 2022). Besides, other non-climatic
factors such as station relocations could also introduce some systematic errors in SSD.
Taking into account the aforementioned issues in SSD observations, it is imperative to
detect and adjust the discontinuities of SSD series in China, especially in the presence of
artificial errors caused by changes in observing instruments, station locations, nearby
environmental conditions, observing procedures, or other factors. To achieve this, this study
compiled raw SSD data and systematically corrected the known day0-type discontinuity, as



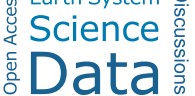

described in Section 2.1-2.2. In Section 2.3, a homogenization procedure was described to
detect and adjust series discontinuities in the variance and mean of daily SSD, with establishing
a reliable reference series. Section 3.1-3.2 analyzed the detected breakpoints and assessed the
impacts of series homogenization on trends across various periods. We further examined the
influence of cloud cover and aerosols on SSD variations in China in Section 3.3. This study
produced a 62-year (1961-2022) homogenized daily SSD dataset in China, which are publicly
accessible to support research on China's dimming and brightening phenomena, to improve the
assessment of solar radiation simulations and future projections, and to provide valuable data
for various applications such as solar energy layout.

**2. Data and methods**
**2.1 Data**

The daily observed SSD at 2,425 meteorological stations from 1961 to 2022 were collected

from the China Meteorological Administration (CMA, http://data.cma.cn/en/). After screening
stations based on data continuity and length, i.e., ≥15 days of data per month, ≥10 months per
year, and ≥50 years during the entire period, a total of 2,263 stations were involved in this
study. Again, a widespread replacement of instruments across China occurred around 2019,
transitioning from dark-tube sunshine recorder to photoelectric digital sunshine recorder (Wang
et al., 2020).

$R_s$ is highly correlated with SSD, but serious concerns have been raised about the reliability

of observational $R_s$ data due to poor spatial representativeness, temporal discontinuity, and the
effects of urbanization (Wild et al., 2005; Wang et al., 2014; He et al., 2018). In particular for
China, issues related to instrument aging, sensitivity drift, and instrument replacements have
notably contributed to spurious variations in $R_s$ observations at ~100 stations (He and Wang,
2020). Therefore, $R_s$ series from nearby stations are insufficient to serve as reference series
during homogenization due to their sparse distribution and data inhomogeneity. Reanalysis
products have dynamically consistent and spatiotemporally complete atmospheric fields with
high resolution and open access of data, addressing these limitations of $R_s$ observations (Zhou



et al., 2017). Among these reanalysis products, ERA5 has been verified to outperform in $R_s$
simulations across hourly, daily, monthly, interannual, and decadal scales in China (He et al.,
2021; Li et al., 2023). Leveraging the ERA5 $R_s$ data
(https://cds.climate.copernicus.eu/cdsapp#!/dataset/reanalysis-era5-single-levels), we
estimated sunshine duration based on the criteria of hourly direct $R_s$ exceeding 120 W/m² that
is consistent with instrument measurements (He et al., 2018), to serve as a reference series for
homogenizing the observational SSD data. Meanwhile, SSD estimate from hourly direct $R_s$ of
MERRA2 data from 1980 to 2022 (https://gmao.gsfc.nasa.gov/reanalysis/MERRA-2/) was
used as an aid in constructing the reference series.

To examine the effects of cloud and aerosol on SSD variations, daily cloud cover fraction

(CCF) and aerosol optical depth at 550nm (AOD) from 2003 to 2022 were obtained from a
MODIS product at 1°×1° grids (MCD06COSP_D3,
https://ladsweb.modaps.eosdis.nasa.gov/archive/allData/62/MCD06COSP_D3_MODIS)
(Pincus et al., 2012; Swales et al., 2018).

According to topography and administrative divisions in China, seven subregions were

identified as shown in Figure 1a, i.e., Northwest China (I), the Tibetan Plateau (II), Southwest
China (III), Northeast China (IV), North China (V), Southeast China (VI), and the Loess
Plateau (VII). A relocation event of station was defined as Δlatitude > 0.01°, Δlongitude >
0.01°, or Δaltitude > 10m before and after a specific date. The history of station relocations in
China is reflected by the number of relocations in Figure 1a, and the fraction of the stations
with relocations from 1961 to 2022 in Figure 1b. The average of the number of relocations in
China is about 4 (Fig. 1a). The instrument replacements in 2019 are accompanied with changes
of the measurement height, presenting the unusually frequent relocations at the time (Fig. 1b).
**2.2 Correction of known day0-type discontinuity**

Following the update of automatic sunshine recorders around 2019, which improved

measurement precision from 0.1 hour to 1 minute (Lang et al., 2021), we observed a sudden
reduction in the frequency of zero SSD at specific stations in China after 2019. In most
instances, raw daily SSD is absent of a value of zero for more than six consecutive months,



which is significantly different from series pattern observed prior to 2019. We identified the
segments with this known day0-type discontinuity that characterized by more than six
consecutive months of non-zero SSD.
Results show that the day0-type discontinuity occurs almost in one segment per station,
totaling 378 stations (i.e., 17% of stations in China) distributed mainly across northern China,
Tibetan Plateau, and part of Southwest China (Fig. 2a), and is concentrated in 2019 to 2020
(Fig. 2b). Note that the improved precision may not lead to a notable day0-type discontinuity
in some regions or such minor discontinuities may not be easily identifiable. The spatial
distribution of discontinuities and the years of their most frequent occurrence align with the
update of automatic sunshine recorders in 2019 or later, as well as with station relocations
(Figs. 1 vs 2). We employed the quantile-matching (QM) algorithm to correct the segments
with the identified day0-type discontinuities by utilizing the longest segment that is free of the
discontinuity, which produced the SSD0 series for the subsequent homogenization. The
magnitude of correction reaches up to -5 hours·day$^{-1}$ at two example stations in Northeast
China and Northwest China, respectively (Fig. 3a and 3d). After correction, the frequency of
low values, especially zero values, has increased visibly in 2019 or later in the SSD0 series
compared to the raw SSD series, and then is comparable to the frequency before 2019 (Fig. 3b
and 3e). The monthly SSD0 anomaly after correction appears to be more continuous (Fig. 3c
and 3f). Note that the mean shift of the segment after 2020 like at station No. 51627 (Fig. 3f)
would be statistically homogenized in the following sections.
**2.3 Homogenization procedure**
Since parallel observations for the photoelectric digital SSD recorder and manual SSD
recorder are not publicly available, we are unable to directly explore the relationship between
the two datasets. Data series homogenization offers us with an effective way to address
discontinuities in climate time series caused by non-climatic factors like station relocation and
instrument replacements. Much effort has been devoted to develop homogenization methods,
such as the standard normal homogeneity test (SNHT) (Alexandersson, 1986), two-phase
regression-based methods (Solow, 1987), Bayesian-based methods (Perreault et al., 2000; Chu
and Zhao, 2004), penalized maximal $T$ test (PMT) (Wang et al., 2007), and penalized maximal



*F* test (PMF) (Wang, 2008a). Reeves et al. (2007) compared these methods and argued that
SNH test may work best when trend and periodic effects are diminished by using homogeneous
reference series. However, Wang et al. (2007) and Wang (2008a) pointed out that unequal
sample sizes affect the false alarm rate and detection power of SNHT-type tests. They
demonstrated that PMT and PMF tests with incorporating penalized empirical corrections offer
higher detection power and are suited for long-term series with significant climate trends
(Wang, 2008b; Wang et al., 2010). Since their release, PMT and PMF tests have been
successfully applied to various climate elements including temperature, precipitation,
humidity, wind speed, and $R_s$ (Wang et al., 2010; Dai et al., 2011; Domonkos, 2011; Yang et
al., 2018; Zhou et al., 2018; Ma et al., 2022; Zhou et al., 2022), making them the chosen
methods for this study.
During the homogenization, a well-established reference series is essential for sufficiently
detecting and adjusting inhomogeneities in long-term climate time series, since it can help
remove most real climate changes and synoptic variations (i.e., noise), thereby improving the
signal-to-noise ratio of discontinuities and enabling statistical detection and removal of
spurious shifts (Dai et al., 2011; Zhou et al., 2022). In this study, we first established a reliable
reference series to account for background weather and climate variations, and then detected
and adjusted spurious breakpoints in the mean and variance of the non-zero daily SSD0 series
using the well-established ERA5 reference series, resulting in a homogenized daily SSD
observational dataset.
**2.3.1 Construction of the reference series**
A reliable reference series should effectively capture most background weather and climate
variations while remaining homogeneous. ERA5 SSD series is highly correlated with the SSD0
series on daily and monthly time scales across China (Fig. 4a and 4d), which ensures that ERA5
SSD as a reference series can remove most background weather and climate variations from
the SSD0 series, thereby facilitating the detection of breakpoints.
Previous studies have indicated that ERA5 significantly overestimates the variation in $R_s$
from 2003 to 2010 in China (He et al., 2021; Shao et al., 2022). This overestimation is inherited
in the SSD estimated from hourly direct $R_s$ of ERA5, presenting inhomogeneities during this



period, particular in North China and Southeast China (Fig. 5). We evaluated several reanalysis
products and found that the SSD estimated from hourly direct $R_s$ of MERRA2 does not suffer
from this issue (Fig. 5), maybe since MERRA2 assimilates space-based observations of
aerosols and improves $R_s$ simulations in China to some extent (Feng and Wang, 2021).
Meanwhile, MERRA2 SSD is also highly correlated with the SSD0 series (Fig. 4b and 4e). To
mitigate the discontinuity of ERA5 SSD from 2003 to 2010, we took MERRA2 SSD as the
reference series and applied the PMT test to detect breakpoints in the monthly ERA5 SSD
series. After obtaining the breakpoints, we employed the QM algorithm to adjust discontinuities
in the daily ERA5 SSD series, using the longest segment as the reference. Results show that
the homogenized ERA5 SSD not only exhibits higher correlations with the SSD0 series on
daily and monthly time scales (Fig. 4c and 4f), but also greatly alleviated the overestimation
from 2003 to 2010 (Fig. 5), which makes it a suitable reference series for the subsequent
homogenization.
**2.3.2 Detection and adjustment of breakpoints in non-zero daily SSD0 series**
Zero values in a daily meteorological series should remain unaltered unless supported by
evidence or reports of trace occurrence or changes in measuring precision (Wang et al., 2010).
After the known day0-type discontinuity has been corrected in Section 2.2 above, the
subsequent homogenization was performed on the variance and mean of non-zero daily SSD0
series. To achieve this, we decomposed the non-zero daily SSD0 series into two components:
intramonthly and monthly.
Firstly, we applied an improved Kolmogorov–Smirnov (K-S) test (Dai et al., 2011; Zhou
et al., 2021a) at a 99.9% significance level to the intramonthly component of the daily
difference series ($DSSDd_{intra}$) for detecting breakpoints in the variance of the non-zero daily
SSD0 series:
$$DSSDd_{intra} = SSDa_{obs} - \alpha \cdot SSDa_{ERA5} \qquad (1)$$
$$SSDa_{obs} = SSDd_{obs} - SSDm_{obs} \qquad (2)$$
$$SSDa_{ERA5} = SSDd_{ERA5} - SSDm_{ERA5} \qquad (3)$$
where $SSDd_{obs}$ and $SSDd_{ERA5}$ are the non-zero daily values of SSD0 and homogenized ERA5





SSD; $SSDm_{obs}$ and $SSDm_{ERA5}$ are the monthly mean $SSDd_{obs}$ and $SSDd_{ERA5}$; $SSDa_{obs}$ and
$SSDa_{ERA5}$ are the daily anomalies of $SSDd_{obs}$ and $SSDd_{ERA5}$, respectively; $\alpha$ is the liner
regression coefficient of $SSDa_{obs}$ against $SSDa_{ERA5}$. Systematic biases in the reanalysis and the
effect of the station-versus-grid discrepancies can be greatly eliminated by the regression
against the observation.
Secondly, we applied both PMT and PMF tests developed by Wang et al. (2007) and Wang
(2008b) at a 99% significance level to detect breakpoints in the monthly mean of the non-zero
SSD0 series. The breakpoints detected by both methods within one year were kept. In the PMT
test, $SSDm_{ERA5}$ was taken as the input of reference series. Consistently, the monthly difference
series (DSSDm) requested in the PMF test was constructed as follows:
$$DSSDm = SSDm_{obs} - \beta \cdot SSDm_{ERA5} \qquad (4)$$
where $\beta$ is liner regression coefficient between $SSDm_{obs}$ and $SSDm_{ERA5}$.
To obtain a manageable number of breakpoints in the final, we followed the approach of
Zhou et al. (2021a) by setting 365 days between breakpoints as the threshold to merge the
detected breakpoints above. For cases with three or more breakpoints within 365 days of each
other, we retained only the middle breakpoint, and for two breakpoints, we kept the one with
the larger test statistic.
Finally, we adopted the QM algorithm from Wang et al. (2010) to remove the merged
breakpoints in the daily difference series (DSSDd) that is the residual series from the regression
(the slope $\gamma$) of $SSDd_{obs}$ on $SSDd_{ERA5}$:
$$DSSDd = SSDd_{obs} - \gamma \cdot SSDd_{ERA5} \qquad (5)$$
This produced a homogenized daily SSD dataset for China, covering the period from 1961 to
2022, with zero values backfilled. The longest segment was chosen as the baseline segment
primarily due to its relative homogeneity and reliability. Despite the use of advanced automated
instruments with higher precision from 2019 onwards, the segment is still too short to fully
meet the criteria for a reliable baseline segment. The segment after 2019 will be considered as
the baseline segment when the dataset is updated in the coming years.



## 3. Results

### 3.1 Detection and adjustment of breakpoints

One or two breakpoints in the variance of the DSSDd$_{intra}$ series were detected at 328 stations, mainly in Northwest China, Northeast China, North China, and the Loess Plateau (Fig. 6a). Most of these breakpoints occur around 2019 (Fig. 6a2), coinciding with the instrument replacements from dark-tube sunshine recorder to photoelectric digital sunshine recorder, as well as the station relocations (Fig. 1b). During the period of 1961-2022, 1,238 stations (55%) in China suffered from the breakpoints in the mean of the DSSDm series (Fig. 6b1). These breakpoints are evenly distributed across China (Fig. 6b1), with many occurring around 2019 and two additional small peaks around 1972 and 2003 (Fig. 6b2). Approximately 52% of the stations in China were detected with one breakpoint, 32% with two breakpoints, 12% with three breakpoints, and few stations with more than four breakpoints (Fig. 6b1). After merging the two types of breakpoints above, a total of 1,363 stations experienced breakpoints, accounting for approximately 60% of the stations in China (Fig. 6c1). The merged breakpoints are densely concentrated in northern China, where approximately 71% of the stations are affected, with the highest density (74%) observed on the Loess Plateau, while approximately 47% of the stations are affected in southern China (Fig. 6c1). A higher fraction of stations with breakpoints occurs around 2019 after merging (Fig. 6c2).

The detected breakpoints may be associated with factors such as instrument replacements, station relocations, equipment malfunctions, operation errors, and other environment changes, any of which may contribute to data series inhomogeneity (Sanchez-Lorenzo and Wild, 2012; Wang et al., 2020). To empirically demonstrate it, we attempted to collect such types of information but were only able to compile a detailed set of information about station relocation. We found that over 50% of stations in 2019 were relocated (Fig. 1b), mostly because they needed to change their positions or heights due to instrument replacements or urbanization. The hit probability for matching detected breakpoints with stations relocations is approximately 65%. Noted that the breakpoints may be caused by factors other than station relocations, while some station locations may not have resulted in any breakpoints, or certain breakpoints may not have emerged from the background weather or climate variations that was





not easily detected by a statistical method.
To remove the artificial breakpoints detected above, the QM algorithm was implemented
to achieve homogenization by aligning the empirical distributions of all segments. For
examples, three breakpoints in the variance and mean of the SSD0 series were detected at
Station No. 51627 (Fig. 7). The first two breakpoints are associated with the station relocations
and the last breakpoints are related to the replacement of instrument in 2019. The adjustments
estimated based on the QM algorithm for the three breakpoints are approximately 1, -0.5, and
2.8 hours·day$^{-1}$, respectively (Fig. 7b). After adjustments, the monthly SSD0 anomaly series
appears continuous and reasonable, particularly after 2019 (Fig. 7c).
**3.2 Comparison of trends before and after homogenization**
The discontinuities hidden in the series are bound to affect the estimate of long-term trends
of SSD. Fig. 8 shows series comparisons among the raw SSD, SSD0, and homogenized SSD
averaged over China and its seven subregions. The most significant adjustments are evident in
2019 or later, occurring across China (Fig. 8). This is jointly resulted from two aspects: the
high robustness of dark-tube sunshine recorder in measuring SSD before 2019, and the
widespread switch to photoelectric digital sunshine recorder in 2019 that caused notable shifts
compared to earlier period. Based on the periods of dimming and brightening in China revealed
by prior researches (He et al., 2018; He and Wang, 2020), trend analysis was conducted for two
major periods: 1961-1990 and 1991-2022.
During the period of 1961-1990, the homogenized SSD exhibits a significant downward
trend of -0.11 hours·day$^{-1}$/decade ($p<0.05$) in China, compared to a slightly steeper decline of
-0.13 hours·day$^{-1}$/decade ($p<0.05$) in the raw SSD (Figs. 8-9 and Table 1), though the difference
between the two is not evident. After homogenization, the dimming of homogenized SSD
weakens across China except the Tibetan Plateau, with the most pronounced weakening in
North China by 0.04 hours·day$^{-1}$/decade compared to the raw SSD (Figs. 8-9 and Table 1).
Meanwhile, the Tibetan Plateau shows a reduced and non-significant increase in the
homogenized SSD (0.02 hours·day$^{-1}$/decade with a reduction of 60%, $p>0.10$) compared to the
raw SSD (0.05 hours·day$^{-1}$/decade, $p<0.10$) during the dimming period of China (i.e., 1961 to
1990) (Figs. 8-9 and Table 1), suggesting that the homogenized SSD tends to better describe



the dimming phenomenon.

During the period of 1991-2022, only the southern regions of China experienced slight

brightening, whereas the northern regions continued dimming (Fig. 9). The national average
SSD trend of China remains unchanged before and after homogenization, with a decline of
about -0.04 hours·day$^{-1}$/decade ($p$<0.10) (Table 1). However, the magnitudes of decadal trends
change significantly across various regions. In heavily polluted regions such as North China
and the Loess Plateau, the degree of dimming diminishes in homogenized SSD. Specifically,
the SSD trend decreases from -0.14 hours·day$^{-1}$/decade ($p$<0.05) to -0.12 hours·day$^{-1}$/decade
($p$<0.05) in North China, and from -0.11 hours·day$^{-1}$/decade ($p$<0.05) to -0.08 hours·day$^{-}$
$^{1}$/decade ($p$>0.10) in the Loess Plateau (Figs. 8-9 and Table 1). In addition, for the Tibetan
Plateau and Northeast China, the SSD after homogenization presents a more pronounced
decline compared to the raw data (Figs. 8-9 and Table 1).

In 2013, China issued and implemented the Air Pollution Prevention and Control Action

Plan (APPCAP), to address severe air pollution and its associated health risks. The subsequent
strengthening of air quality measures may have contributed to a reversal of SSD trend
thereafter. During the period of 2013-2022, the national average SSD in China shifts from a
decrease of -0.02 hours·day$^{-1}$/decade to an increase of 0.07 hours·day$^{-1}$/decade after
homogenization, reflecting well the effect of the APPCAP implementation on the SSD trend
reversal compared to earlier periods (Figs. 8-9 and Table 1). Especially for heavily polluted
regions like North China, Southeast China, Loess Plateau, and Northeast China, the
homogenized SSD shows more brightening after homogenization, with the most notable
increase in North China where the trend increases from 0.16 ($p$>0.10) to 0.42 ($p$<0.10)
hours·day$^{-1}$/decade (Figs. 8-9 and Table 1). Due to the instrument replacement in 2019, the
artificial breakpoints in the SSD series have been removed and the homogenized SSD series
appear more continuous to the naked eye (Fig. 8b-8e). Specifically, the homogenized SSD has
a weakened trend (-0.16 hours·day$^{-1}$/decade, $p$>0.10) compared to the raw data (-0.52
hours·day$^{-1}$/decade, $p$<0.05) in Northwest China. The homogenized SSD declines with -0.36
hours·day$^{-1}$/decade ($p$<0.10) and -0.16 hours·day$^{-1}$/decade ($p$>0.10) in the Tibetan Plateau and
Southwest China, respectively (Fig. 8b-d). In summary, considering the uncertainties brought

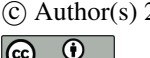


by the series inhomogeneities caused by non-climatic factors such as instrument replacements
and station relocations, it is very necessary to address these inhomogeneities, particularly in
studies focused on detecting and attributing global diming and brightening.
**3.3 Relationships of cloud and aerosol with sunshine duration**
Cloud and aerosol affect the amount of solar radiation reaching the Earth's surface through
sunlight reflection, absorption, and scattering, making their combined effects on solar radiation
a key factor in understanding global dimming and brightening (Wild, 2009; Wild, 2012; Feng
and Wang, 2021; Ma et al., 2022). SSD serves as a core indicator of solar radiation, which is
modulated by both cloud cover and aerosols. Due to limitations of satellite data, this section
focuses on the relationships of cloud cover fraction (CCF) and aerosol optical depth (AOD) on
SSD variations solely over the 20-year period starting from 2003.
Fig. 10 shows maps of decadal changes in AOD, CCF, and the homogenized SSD, and
their time series at the locations collocated with stations in China. For the entire period of 2003-
2022, the correlation coefficient of the averaged CCF in China against the raw SSD is -0.53
($p<0.05$), and its coefficient against the homogenized SSD reaches -0.71 ($p<0.05$). On the other
hand, in the heavily polluted regions such as North China and Northeast China, the correlation
coefficient between AOD and SSD is significantly negative, and the relationships are
intensified after homogenization, i.e., from -0.40 ($p<0.10$) to -0.56 ($p<0.05$) and from -0.41
($p<0.10$) to -0.54 ($p<0.10$), respectively. These relationship changes indicate a stronger
relationship with CCF and AOD in the homogenized SSD dataset.
During the period of 2003-2012, the average SSD in China decreases at a rate of -0.20
hours·day$^{-1}$/decade ($p>0.10$), accompanied by slight increases in both CCF and AOD (Fig.
10g). For regional details, the significant increase of AOD in North China (Fig. 10a) and the
significant increase in CCF in Southeast China (Fig. 10b) jointly contributes to regional
divergences in the SSD decadal changes of China during this period (Fig. 10c).
The effect of the APPCAP implementation on AOD can be clearly seen with a rapid
reduction after 2013 (Fig. 10a, 10d, and 10g). CCF also exhibits a corresponding shift from the
perspective of spatial distribution of its decadal changes, especially in North China (Fig. 10b



vs 10e), maybe due to various cloud-aerosol interactions. These changes of AOD and CCF
contributes to the SSD brightening after 2013, which is reflected in the maps and time series
of their decadal changes (Fig. 10). During the period of 2013-2022, the spatially coherent
pattern of AOD decadal reductions (-0.12 1/decade, $p<0.05$) inevitably lead to an overall SSD
brightening, on the basis of which the spatial detail of CCF decadal changes further inversely
shapes the pattern of SSD decadal changes (Fig. 10d-10f). This results in a decadal change of
0.07 hours·day$^{-1}$/decade ($p>0.10$) in the national average SSD (Fig. 10g). In heavily polluted
regions such as North China, it's clear that decreases in both AOD and CCF jointly result in
the enhanced brightening in the localized SSD (Fig. 10d-10f).

**4. Data availability**
The homogenized dataset of daily sunshine duration at 2.0×2.0 grids in China from 1961
to 2022 generated in this study, provides a valuable database for assessing and attributing solar
radiation variations, and will also be a key for various applications in the solar energy industry,
agricultural management, and ecology and climatology research. The homogenized dataset is
freely accessed via the following link, https://doi.org/10.11888/Atmos.tpdc.301478 (He et al.,

2024).


**5. Conclusions and discussion**
Inhomogeneities in climate series, stemming from non-climatic factors such as instrument
replacements and station relocations, inevitably affect the estimate of long-term trends. While
dark-tube sunshine recorder robustly measured SSD in China prior to 2019, the widespread
transition to photoelectric digital sunshine recorder in 2019 introduced significant non-climate
discontinuities in SSD. After compiling raw SSD observational data, we first noted a sudden
reduction in the frequency of zero SSD in segments from 2019 onwards, attributed to improved
measurement precision from 0.1 hour to 1 minute following the instrument update in 2019.
This known day0-type discontinuity affected a total of 378 stations (~17% of stations in China),
occurring almost in one segment per station, predominantly located in northern China, the



Tibetan Plateau, and parts of Southwest China. We applied the quantile-matching algorithm to
correct the segments with the day0-type discontinuities, using the longest segment unaffected
by the discontinuity, which produced the SSD0 series that has comparable frequencies of zero
SSD before and after 2019.
To further address the remaining discontinuities, we developed a homogenization
procedure for producing a 62-year (1961-2022) homogenized daily SSD dataset in China. First,
a well-established ERA5 SSD was constructed as a reliable reference series with the help of
MERRA2 SSD to eliminate the background weather and climate variations (i.e., noise) for
enhancing the signal-to-noise of artificial discontinuities. Second, two separate steps were
taken to statistically detect discontinuities in the variance and mean of the non-zero daily SSD0
series. Results show that breakpoints in the variance are mainly concentrated in the northern
part of China, while the breakpoints in the mean are evenly distributed across China. After
merging the two types of breakpoints above, a total of 1,363 stations experienced breakpoints,
accounting for ~60% of the stations in China. The peak in the number of breakpoints occurs in
2019, coinciding with the nationwide transition from manual to automated SSD recorders. In
all, ~65% of the detected breakpoints were confirmed by station relocations and associated
instrument replacements. Finally, the merged breakpoints were removed by the quantile-
matching algorithm to produce the final homogenized daily dataset.
Compared to the raw SSD, the homogenized SSD shows more continuous variations across
various time scales, providing a solid basis for estimating reliable long-term trends for various
periods. During the dimming (1961 to 1990), the homogenized SSD presents weakened
dimming across China compared to the raw SSD, particularly in the Tibetan Plateau, where the
trend shifts from a significantly positive to a non-significant negative with a reduction of 60%,
suggesting that the homogenized SSD tends to better describe the dimming phenomenon.
During the period of 1991-2022, only the southern regions of China experienced slight
brightening, whereas the northern regions continued dimming. In heavily polluted regions such
as North China and the Loess Plateau, the extent of dimming diminishes in homogenized SSD.
The subsequent strengthening of air quality measures after issuing the APPCAP in 2013 in
China may have contributed to a reversal of SSD trend thereafter. During the period of 2013-



2022, the national average SSD in China shifts from a decrease of -0.02 hours·day$^{-1}$/decade to
an increase of 0.07 hours·day$^{-1}$/decade after homogenization, reflecting well the effect of the
APPCAP implementation on the SSD trend reversal compared to earlier periods. Especially in
heavily polluted regions, the homogenized SSD shows more brightening after homogenization,
with the most notable increase observed in North China.
We further examined the regulatory effects of clouds and aerosols on SSD changes using
the satellite data from 2003 to 2022. Our analysis reveals that the relationships of CCF and
AOD with SSD are intensified in the homogenized dataset. During the period of 2003-2012,
the average SSD in China decreases at a rate of -0.20 hours·day$^{-1}$/decade ($p$>0.10),
accompanied by slight increases in both CCF and AOD. The effect of the APPCAP
implementation on AOD is evident, with a rapid reduction in AOD after 2013. In the
subsequent period from 2013 to 2022, the spatially coherent pattern of AOD decadal reductions
results in an overall SSD brightening, on the basis of which the spatial detail of CCF decadal
changes further inversely shapes the pattern of SSD decadal changes. This leads to a national
average SSD change of 0.07 hours·day$^{-1}$/decade ($p$>0.10). These regulatory effects of clouds
and aerosols on SSD obtained using only 20-years satellite CCF and AOD data were also
confirmed by prior studies regarding $R_s$. For instances, earlier studies have demonstrated that
clouds were only able to explain $R_s$ changes in the southern part of China before 1990, but
accounted for changes across the entire China after 1990 (Yang et al., 2013; He and Wang,
2020). Wang et al. (2012) suggested that seasonal and interannual variations in $R_s$ are
predominantly affected by clouds, while decadal variations are mainly dominated by aerosols.
Our long-term homogenized daily SSD dataset not only enables a reliable assessment of
global dimming and brightening in China, but also provides valuable insights for planning,
designing, and evaluating benefits across various sectors, including solar energy, agriculture,
and environmental management. Moreover, the homogenization experience including
constructing a reference series with the aid of current atmospheric reanalysis could be adapted
to have broader applications in the homogenization of other climate elements or over other
regions to develop a global dataset.



**Author contributions**
YH, KW, and KY designed the research. YH performed the analysis and wrote the draft. CZ
advised the homogenization method. KW and CY collected raw SSD observational data and
YH compiled all the remaining data. All the authors jointly contributed to interpreting the
results and editing the final paper.
**Competing interests**
The authors have declared no any competing interests.

**Acknowledgements** This study was supported by the National Natural Science Foundation of
China (No. 42205171), the Starting Grant for Introduced Talents of Sun Yat-sen University
(37000-12230039), and the Central University Basic Research Business Fee on the Training of
Young Teachers (37000-31610006). Sunshine duration and other meteorological observations
at 2,425 stations were obtained from the China Meteorological Administration (CMA,
http://data.cma.cn/en). ERA5 and MEERA2 data were downloaded respectively at
https://www.ecmwf.int/en/forecasts/datasets/reanalysis-datasets/era5 and
https://gmao.gsfc.nasa.gov/reanalysis/MERRA-2/. MODIS data were downloaded at
https://ladsweb.modaps.eosdis.nasa.gov/archive/allData/62/MCD06COSP_D3_MODIS.



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





**Table 1** Trends of sunshine duration (unit: hours·day$^{-1}$/decade) before and after
homogenization in China and its seven subregions, i.e., Northwest China (I), Tibetan Plateau
(II), Southwest China (III), Northeast China (IV), North China (V), Southeast China (VI), and
Loess Plateau (VII), during three periods of 1961-1990, 1991-2022, and 2013-2022. Trends
with a significance level of 0.05 are shown in bold, while those with a significance level of 0.1
are italicized.

| | 1961-1990 | | 1991-2022 | | 2013-2022 | |
|---|---|---|---|---|---|---|
| | Before | After | Before | After | Before | After |
| China | **-0.13** | **-0.11** | *-0.04* | *-0.04* | -0.02 | 0.07 |
| Northwest China (I) | -0.04 | -0.02 | -0.06 | -0.03 | **-0.52** | -0.16 |
| Tibet Plateau (II) | *0.05* | 0.02 | *-0.05* | **-0.07** | -0.20 | *-0.36* |
| Southwest China (III) | **-0.13** | **-0.11** | *0.06* | 0.02 | -0.05 | -0.16 |
| Northeast China (IV) | **-0.11** | **-0.10** | -0.01 | -0.02 | 0.23 | 0.26 |
| North China (V) | **-0.22** | **-0.18** | **-0.14** | **-0.12** | 0.16 | *0.42* |
| Southeast China (VI) | **-0.23** | **-0.21** | -0.02 | -0.02 | 0.10 | 0.21 |
| Loess Plateau (VII) | *-0.13* | -0.11 | **-0.11** | -0.08 | -0.08 | 0.11 |

Data

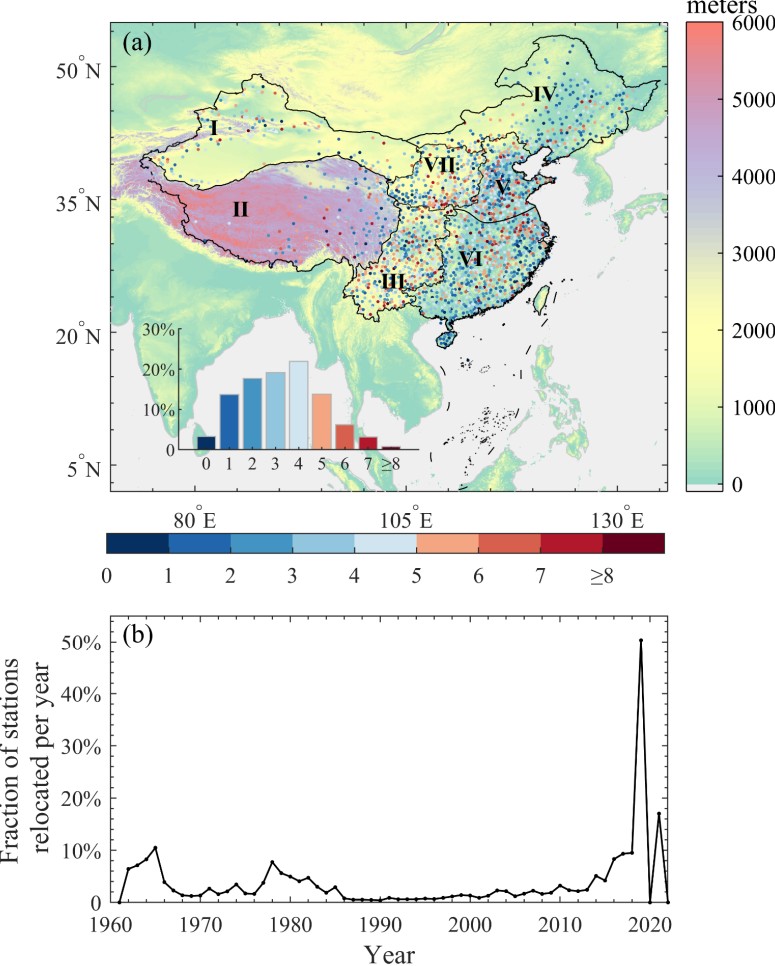


**Figure 1** (a) Map of the number of relocations for 2,263 national meteorological stations (dots colored by the bottom color bar, unit: times) in China during the period of 1961 to 2022. A relocation event is defined as $\Delta$latitude > 0.01°, $\Delta$longitude > 0.01°, or $\Delta$altitude > 10m before and after a specific date. The elevation map serves as the background and is colored by the right-side color bar. The sub-figure in the bottom left illustrates the percentage of stations corresponding to the number of relocations for all stations. According to topography and administrative divisions of China, seven subregions were identified, i.e., Northwest China (I), Tibetan Plateau (II), Southwest China (III), Northeast China (IV), North China (V), Southeast China (VI), and Loess Plateau (VII). (b) Time series of the fraction of stations (unit: %) that underwent one or more relocations per year. The unusually frequent relocations in 2019 were accompanied with the instrument replacements that occurred that year.
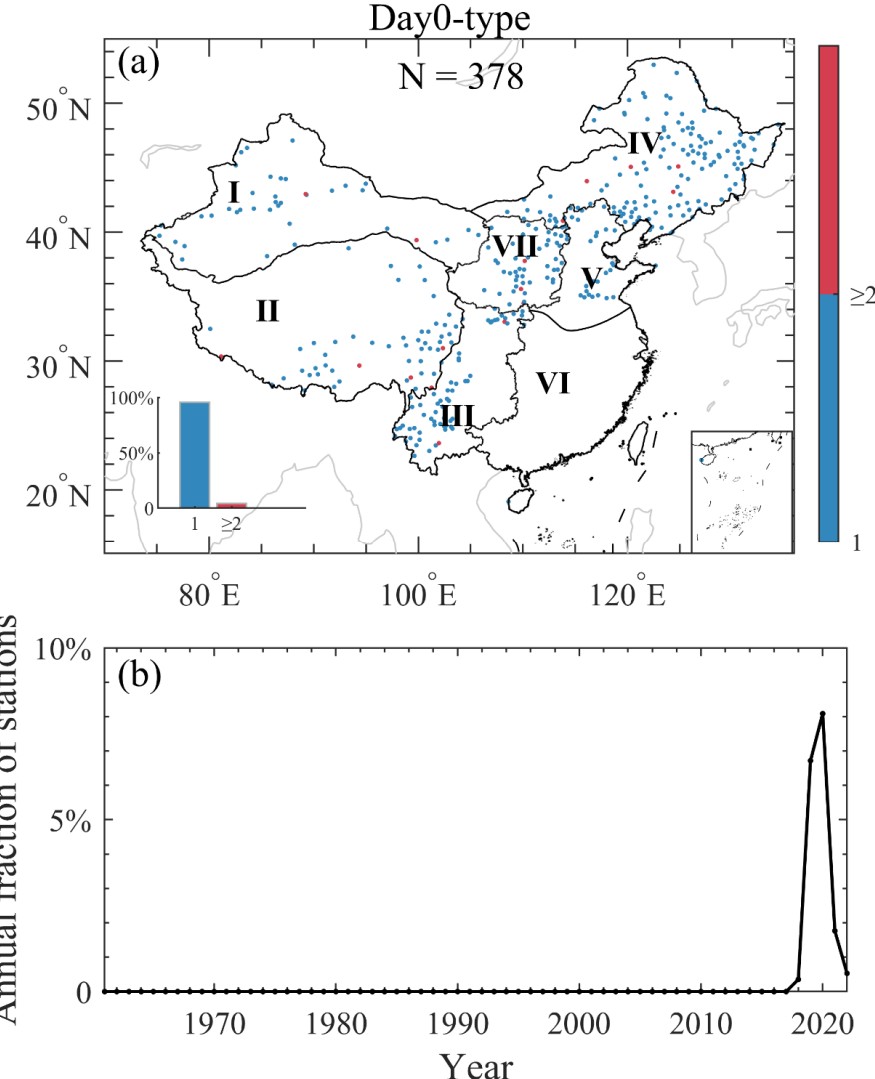

652

**Figure 2** (a) Map of stations with the day0-type discontinuities in the monthly count of days
with zero sunshine duration. The right-side color bar illustrates the total number of segments
with the day0-type discontinuities for each station. The sub-figure in the bottom left shows the
percentage of stations with different numbers of such segments per station. A total of 378
stations were identified with the day0-type discontinuities. (b) Annual fraction (unit: %) of
stations with the day0-type discontinuities from 1961 to 2022.



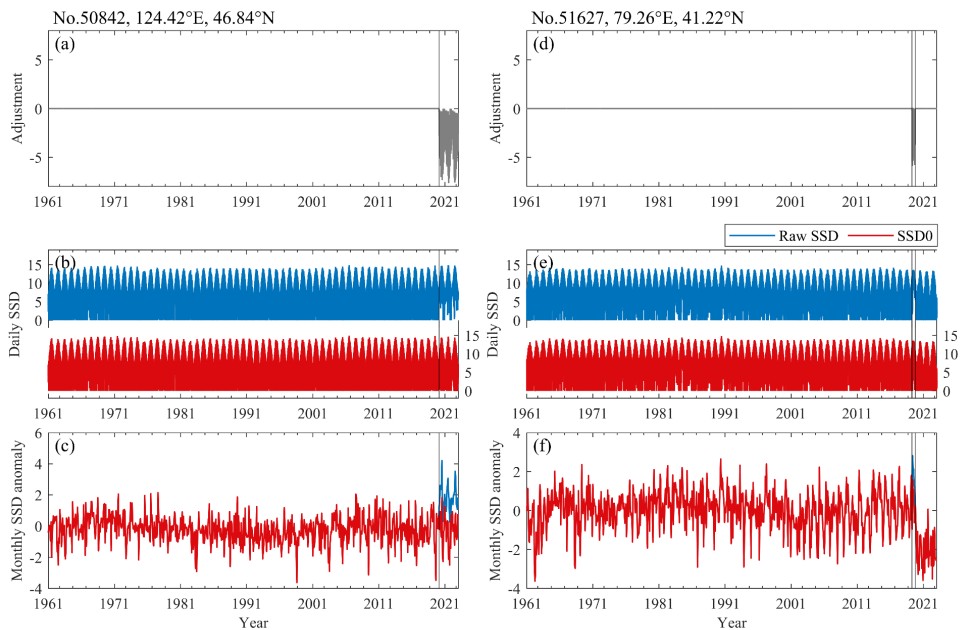

**Figure 3** Comparison of raw sunshine duration (Raw SSD, blue line, unit: hours·day$^{-1}$) with the day0-type corrected sunshine duration (SSD0, red line, unit: hours·day$^{-1}$) at two example stations in (a-c) Northeast China and (d-f) Northwest China, respectively. (a and d) QM adjustments added to the raw SSD; (b and e) Daily time series of the raw SSD and SSD0; (c and f) as in (b and e), but for their monthly SSD anomalies. The vertical grey lines indicate the start and end dates of segments identified with the day0-type discontinuities.





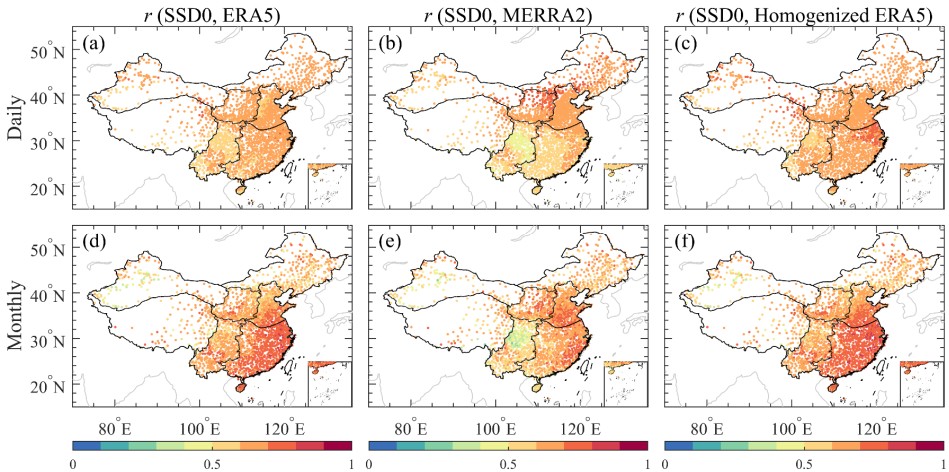


**Figure 4** Maps of the correlation coefficients of sunshine duration at daily, monthly and annual

time scales between the day0-type corrected observation (SSD0) and ERA5, MERRA2 as well

as homogenized ERA5.



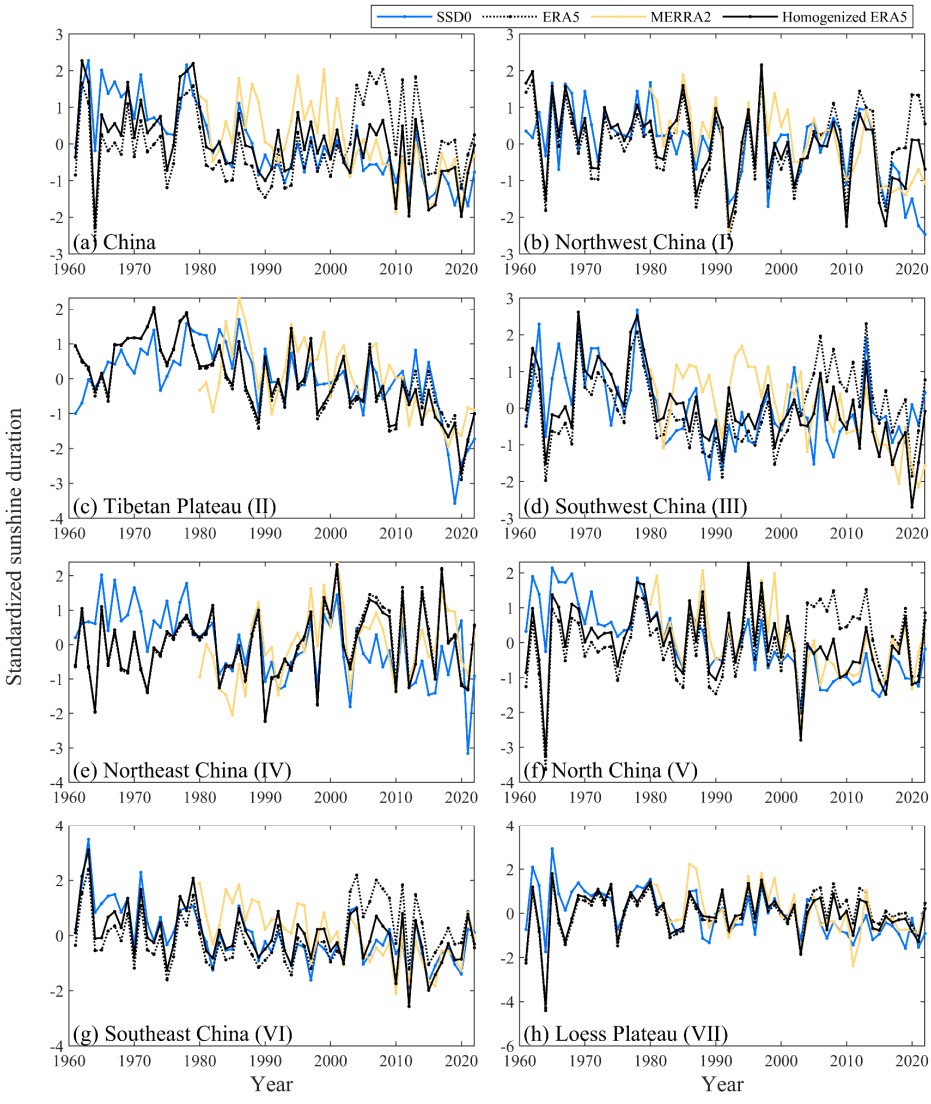


**Figure 5** Time series of the standardized sunshine duration (unit: 1) from the day0-type
corrected observation (SSD0, blue line), ERA5 (black-dotted line), MERRA2 (yellow line),
and homogenized ERA5 (black line) in (a) China and (b-h) its seven subregions.

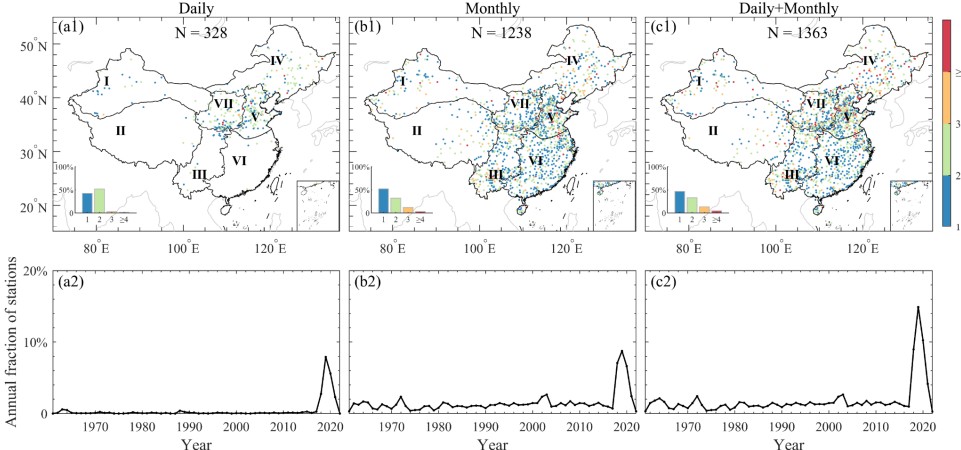

**Figure 6** (a1 and b1) Maps of the number of breakpoints detected in the daily variance and monthly mean of the non-zero SSD0 series, respectively. (c1) Map of the number of breakpoints merged from those in Figure 6a1 and 6b1. The total number (N) of stations with one or more breakpoints from 1961 to 2022 is shown in each panel. (a2, b2, and c2) Annual fraction (unit: %) of stations with the breakpoints from 1961 to 2022.



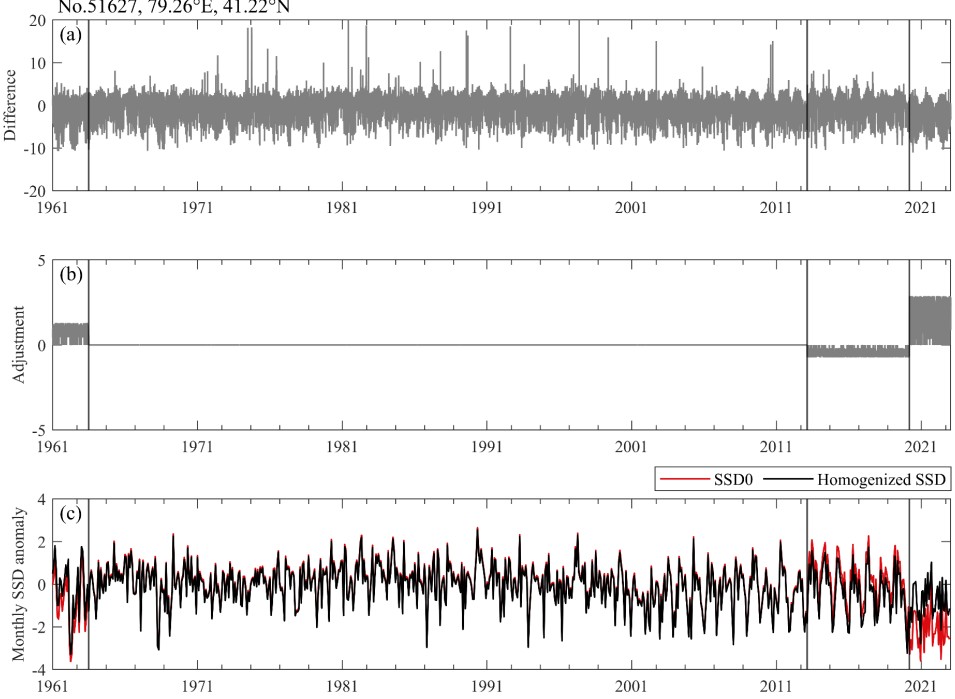

680

**Figure 7** Comparison of the day0-type corrected sunshine duration (SSD0, red line) before and after homogenization (Homogenized SSD, black line) at an example Station No. 51627 in Northwest China. (a) Daily SSD difference (unit: hours·day$^{-1}$) between the SSD0 and the corrected ERA5 reference series; (b) QM adjustments added to the SSD0; (c) Monthly anomaly series of the SSD0 (red line) and homogenized data (black line). The vertical lines indicate the dates of the breakpoints detected.

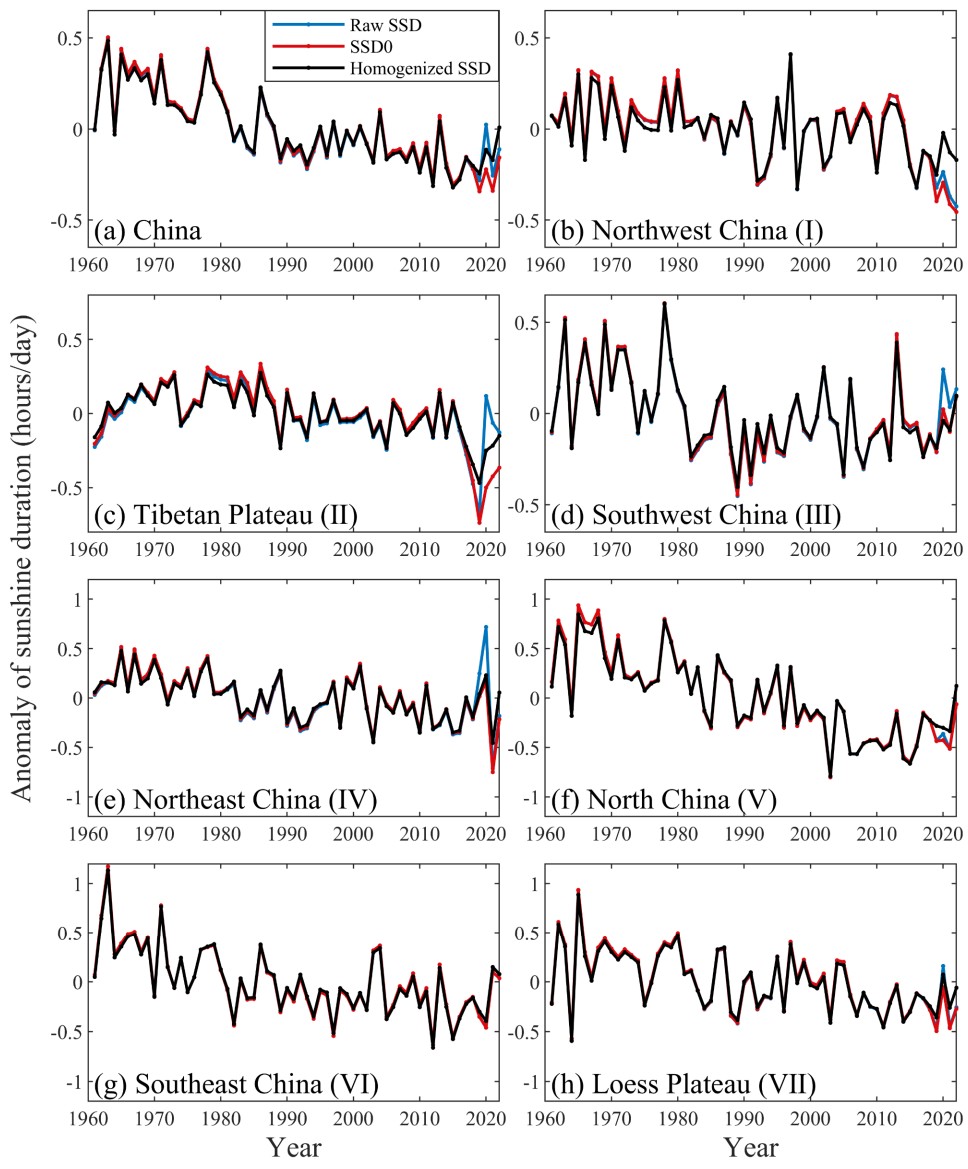

687

**Figure 8** Time series of raw sunshine duration (Raw SSD, blue line), the day0-type corrected

SSD0 (red line), and the homogenized SSD (black line) in (a) China and (b-h) its seven

subregions from 1961 to 2022. The anomaly is referenced to the average for the entire period.

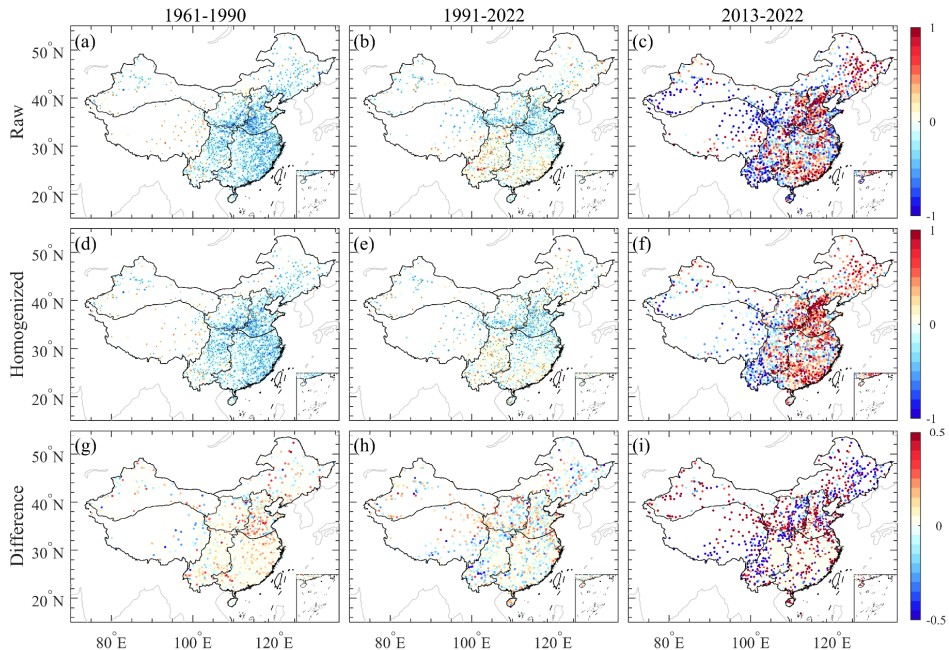

**Figure 9** Maps of the decadal changes (unit: hours·day$^{-1}$/decade) in (a-c) raw sunshine duration, (d-f) homogenized SSD, and (g-i) their difference over China during three periods of 1961-1990, 1991-2022, and 2013-2022, respectively. Black dots superimposed on the colored circles indicate a significance level of 0.10.

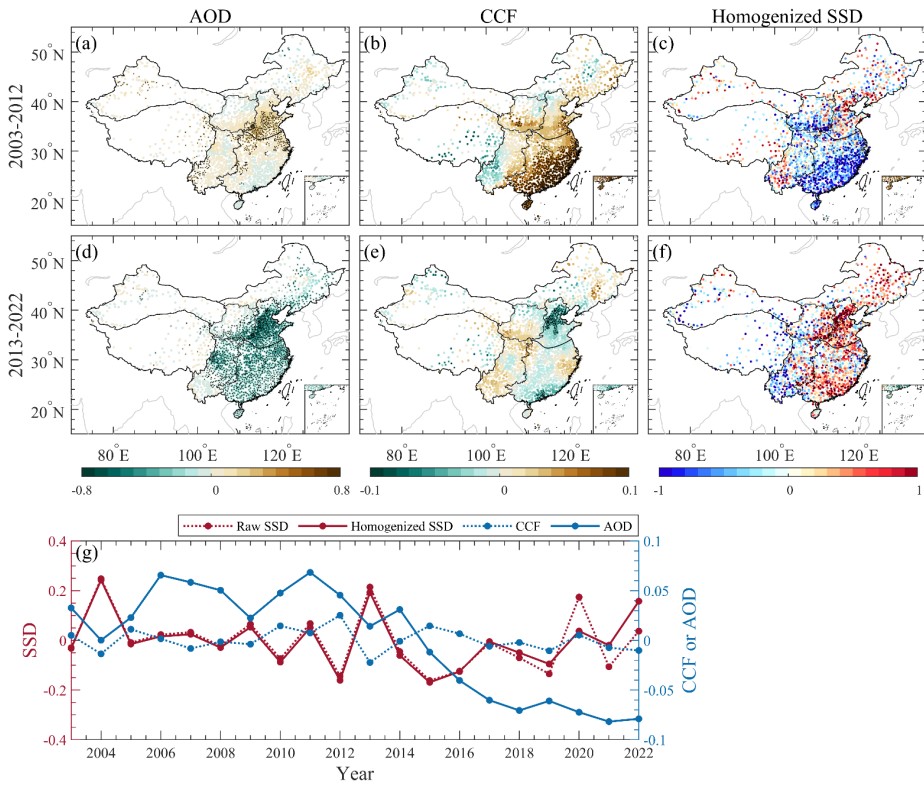

696

**Figure 10** (a-c) Maps of the decadal changes in aerosol optical depth (AOD, unit: 1/decade),
cloud cover fraction (CCF, unit: 1/decade), and homogenized sunshine duration (Homogenized
SSD, unit: hours·day$^{-1}$/decade) over China from 2003 to 2012. (d-f) Same as Figure 10a-10c,
but from 2013 to 2022. Black dots indicate a significance level of 0.10. (g) Time series of the
raw SSD (red dotted line), homogenized SSD (red solid line), CCF (blue dotted line) and AOD
(blue solid line) from 2003 to 2022.