# Peer review of "Yanyi He1,3, Kaicun Wang2\*, Kun Yang3, Chunlüe Zhou1\*, Changkun Shao3, and Changjian Yin2 3 4 1 School of Geography and Planning, Sun Yat-sen University, 510275 Guangzhou, China 5 2 Institute of Carbon Neutrality, Sino-French Institute for Earth System Science, College of 6 7 Urban and Environmental Sciences, Peking University, 100081 Beijing, China 8 3 Department of Earth System Science, Minist"

_Earth System Science Data, 2024_

## Author Comment (AC1)

**Replies to Reviewer Comments (RC1)**

**General Comment:** The manuscript compiles raw SSD observational data from 1961 to 2022 across over 2,200 stations in China, addressing the known day0-type discontinuity post-2019 instrument updates. This contributes significantly to understanding global dimming and brightening phenomena and their implications for solar energy planning and agricultural management. The originality lies in providing a 62-year homogenized daily SSD dataset, which is a valuable resource for the research community. But I am concerned about the following points:

**Response:** Thank you for acknowledging the originality of this study and for your strong recommendation of our manuscript. Based on your constructive comments, we have added some discussion, clarifications, and literature comparisons, and also made further language checks carefully, which greatly improves the readability of the revised manuscript. Below are our point-by-point responses to your concerns.

**Specific Comments:**

**1) Comment:** The study concludes that the homogenized SSD dataset more accurately describes the dimming phenomenon. This is an important finding, but the authors must ensure the statistical significance of this conclusion and further explain its scientific implications in the discussion.

**Response:** In addition to its original inclusion in Section 3.2, we have explicitly highlighted the statistical significance of the dimming captured by the homogenized SSD dataset in Abstract of the revised manuscript: Compared to the raw SSD, the homogenized SSD …, and presents weakened dimming ($p<0.05$) across China from 1961 to 1990 ...

As suggested, we have provided its scientific implication in Section 5 (i.e., Conclusions and discussion): The future use of the homogenized SSD in solar radiation estimation could better correct the spurious largest dimming trend in China during this period revealed by low-quality ground-based observation of solar radiation (Wild, 2012; He et al., 2018; Tang et al., 2023).

**2) Comment:** The manuscript has developed a homogenization procedure to produce a 62-year (1961-2022) homogenized daily SSD dataset in China. However, there is a concern about the incorporation of MERRA2 as a reference series when establishing the ERA5 SSD reference series, which may introduce uncertainties from two different reanalysis data sets. It is essential to explain how these uncertainties are mitigated and their potential impact on the study's results.

**Response:** Thanks for your valuable suggestion. To mitigate uncertainties arising from incorporating other reanalysis products in correcting the overestimation of ERA5 SSD, we first evaluated multiple reanalysis products and identified MERRA2 SSD as the most reliable. Second, the correction is strictly limited to the period of 2003-2010, minimizing its potential impact on the whole study period of 1961-2022. We have also revised such information in Section 2.3.1: We evaluated multiple reanalysis products and found that the SSD estimated from hourly direct $R_s$ of MERRA2 does not suffer from this issue (Fig. 5), …. Note that, to minimize potential uncertainties as much as possible arising from incorporating MERRA2 SSD, the detection and adjustment described above are strictly limited to the period of 2003-2010. Results show that the homogenized ERA5 SSD not only exhibits higher correlations with the SSD0 series on daily and monthly time scales (Fig. 4c and 4f), but also greatly alleviated the overestimation from 2003 to 2010 (Fig. 5), which makes it a suitable reference series for the subsequent homogenization.

**3) Comment:** Some sentences are structurally complex, which may hinder understanding. The authors are encouraged to simplify these sentences to make the article more accessible.

**Response:** Following your suggestions, we have split 9 long sentences and conducted a thorough language check carefully, which greatly increases the readability of our manuscript. For example, the original sentence:

Despite the absence of a sensitivity drift issue in SSD observations by manual sunshine recorders, attributed to the daily replacement of photosensitized paper (Sanchez-Lorenzo and Wild, 2012), the observational data still face challenges in ensuring consistency due to the subjectivity introduced by different observers in practice.

has been revised to:

Manual sunshine recorders do not suffer from a sensitivity drift issue in SSD observations, attributed to the daily replacement of photosensitized paper (Sanchez-Lorenzo and Wild, 2012). However, the consistency of observational data still faces a challenge due to the subjectivity introduced by different observers in practice.

**4) Comment:** Comparison with Other Studies: The authors should discuss how their findings compare with those of other studies. If the results are inconsistent with other research, the reasons for these discrepancies should be explored.

**Response:** As suggested, we have added the comparison with previous studies in Section 5:

in Lines 460-465:

Xia (2010) reported a homogenized SSD trend in China during this period that is consistent with our result, but did not address the day0-type discontinuity issue and only limited the analysis to 2005. The future use of the homogenized SSD in solar radiation estimation could better correct the spurious largest dimming trend in China during this period revealed by low-quality ground-based observation of solar radiation (Wild, 2012; He et al., 2018; Tang et al., 2023).

in Lines 492-494:

It would be valuable to further investigate whether urbanization effect on SSD emerges after the APPCAP implementation in 2013 despite Wang et al. (2017) reporting no such effect before 2013.

---

## Author Comment (AC2)

**Replies to Reviewer Comments (RC2)**

**General Comment:** The authors made significant efforts to integrate observational data of sunshine duration in China, including for the first time to address a sharp drop in zero-value frequency after 2019 caused by the instrument upgrade and to adjust inhomogeneities of its long-term series. This produces the first homogenized daily observational dataset of sunshine duration over China from 1961 to 2022. This effectively addresses critical gaps in data availability and homogeneity of sunshine duration observation providing a crucial dataset to accurately assess the dimming and brightening and to support other practical applications. The manuscript is well-structured, clearly presenting the research goals, statistical methods, dataset description, and results.

The manuscript is recommended for publication with minor revisions as specified below:

**Response:** Thank you for your high recommendation and providing constructive suggestions. Following your suggestions, we have revised a figure, added a new figure, and provided more details on methods and additional explanations, which makes clearer the revised manuscript. Below please find our point-by-point responses to your comments.

**Specific Comments:**

**1) Comment:** It's not clear if the presentation of spatial patterns of regression slopes (α, β, γ in Section 2.3.2) aids to understand the use of reference series during the homogenization. Could you plot them and assess their necessity in the main text?

**Response:** Following your suggestion, we have plotted a new Figure 6 in the main text to show spatial patterns of regression slopes (α, β, γ in Section 2.3.2), which is copied here. Correspondingly, we have added some explanations in Section 2.3.2.

[Figure]

**Figure 6** (a) Map of the linear regression coefficient ($\alpha$) of the daily anomalies of SSDd$_{obs}$ against SSDd$_{ERA5}$. (b-c) Same as (a), but showing $\beta$ for SSDm$_{obs}$ against SSDm$_{ERA5}$, and $\gamma$ for SSDd$_{obs}$ against SSDd$_{ERA5}$, respectively. For more details, refer to Equations 1, 4, and 5.

**2) Comment:** Even though the methods used in the homogenization are widely recognized, a concise description of PMF and PMT test algorithms should be included to help understand the detected breakpoints in Section 2.3. Additionally, more details of the improved K-S test should be added in Section 2.3.1 to enhance its readability.

**Response:** As suggested, we have added more descriptions of the PMF and PMT test algorithms as well as the improved K-S test in Section 2.3. These brief descriptions actually enhance the readability of these methods to some extent.

Added details in Section 2.3: … The PMT test searches for the most likely location of mean shifts in segments of the difference between the candidate and reference series using a recursive testing algorithm (Wang et al., 2007). The PMF test, on the other hand, detects undocumented mean shifts in the difference series with a linear trend by employing a common-trend two-phase regression model (Wang, 2008a). Both test algorithms account for the lag-1 autocorrelation of the series. … For variance shifts in the series, an improved Kolmogorov–Smirnov (K-S) test has been widely used to assess whether two samples follow similar or different distributions (Press et al. 1992). To account for the effects of the lag-1 autocorrelation and sample size, Dai et al. (2011) and Zhou et al. (2021a) developed critical values for given significance levels through a series of Monte Carlo simulations.

**3) Comment:** Provide additional details on daily cloud cover fraction and aerosol optical depth at 500nm (AOD) from MODIS in Section 2.1 to help understand their relationships with sunshine duration in the following text.

**Response:** As suggested, we have provided additional details on cloud cover fraction and aerosol optical depth of MODIS in Section 2.1: Cloud cover fraction is calculated as the percentage of a grid cell that is covered by clouds, with values ranging from 0 (no clouds) to 1 (completely overcast). AOD at 550nm is a measure of the total aerosol content in the atmosphere, quantifying how much sunlight is absorbed and scattered by aerosols, with values ranging from 0 (no aerosol) to values greater than 1 (high aerosol loading).

**4) Comment:** Line 298: Please clarify how to calculate the hit probability of 65%.

**Response:** This information has been clarified in Lines xxx: Since the date of a station relocation does not always align with the date of a statistically detected breakpoint, this value is calculated as the ratio of the number of breakpoints that have one or more relocations within one year of the breakpoint to the total number of breakpoints.

**5) Comment:** The subfigure in Figure 6 is too small to be easily readable. Please revise it.

**Response:** Corrected as suggested.